# Morpho-anatomical determinants of yield potential in *Olea europaea* L. cultivars belonging to diversified origin grown in semi-arid environments

Iftikhar Ahmad[1], Mohammad Sohail[1], Mansoor Hameed[2], Sana Fatima[3], Muhammad Sajid Aqeel Ahmad[2], Farooq Ahmad[2], Ansar Mehmood[4], Sana Basharat[2], Ansa Asghar[2], Syed Mohsan Raza Shah[5], Khawaja Shafique Ahmad[4]*

1 Department of Botany, University of Sargodha, Sargodha, Punjab, Pakistan, 2 Department of Botany, University of Agriculture Faisalabad, Pakistan, 3 Department of Botany, The Government Sadiq College Women University, Bahawalpur, Pakistan, 4 Department of Botany, University of Poonch Rawlakot, Azad Jammu and Kashmir, Pakistan, 5 Department of Botany, Division of Science and Technology, University of Education, Lahore, Pakistan

* ahmadks@upr.edu.pk

## Abstract

Plant performance is mainly estimated based on plant architecture, leaf features and internal microstructural changes. Olive (*Olea europaea* L.) is a drought tolerant, oil yielding, and medium sized woody tree that shows specific structural and functional modifications under changing environment. This study was aimed to know the microstructural alteration involving in growth and yield responses of different Olive cultivars. Eleven cultivars were collected all over the world and were planted at Olive germplasm unit, Barani Agricultural Research Institute, Chakwal (Punjab) Pakistan, during September to November 2017. Plant material was collected to correlate morpho-anatomical traits with yield contributing characteristics. Overall, the studied morphological characters, yield and yield parameters, and root, stem and leaf anatomical features varied highly significantly in all olive cultivars. The most promising cultivar regarding yield was Erlik, in which plant height seed weight and root anatomical characteristics, i.e., epidermal thickness and phloem thickness, stem features like collenchymatous thickness, phloem thickness and metaxylem vessel diameter, and leaf traits like midrib thickness, palisade cell thickness a phloem thickness were the maximum. The second best Hamdi showed the maximum plant height, fruit length, weight and diameter and seed length and weight. It also showed maximum stem phloem thickness, midrib and lamina thicknesses, palisade cell thickness. Fruit yield in the studied olive cultivars can be more closely linked to high proportion of storage parenchyma, broader xylem vessels and phloem proportion, dermal tissue, and high proportion of collenchyma.

**Data Availability Statement:** All relevant data are within the paper and its Supporting Information files.

**Funding:** The author(s) received no specific funding for this work.

**Competing interests:** The authors have declared that no competing interests exist.

## Introduction

The productivity of plants is mostly determined by how well they grow and develop in a given environment. The performance of plants is governed by a number of developmental factors, including plant architecture, leaf characteristics, and internal microstructural changes such dermal, mechanical, vascular, and storage tissues architecture [1,2]. The total amount of light interception, photosynthetic efficiency, and plant strength are determined by the architecture of the plant and its leaves [3]. In plants, dermal and mechanical tissues give soft tissues stiffness in the context of microstructural architecture to prevent collapse [4]. Phloem tissues are involved in photosynthate assimilation from source to sink and in the partitioning of photo-assimilated carbon, and storage tissues (cortex and pith) sustain the vigor and vitality of developing tissues by storing more water in plants [5,6]. These characteristics could therefore be potential contributors to the developmental module that controls crop performance and yield [7]. Subsequently, the optimization of such developmental features seems crucial for high performance of plants [8].

Environmental stress is a crucial factor restricting plant growth as well as crop productivity, thus influencing agricultural sustainability [9]. Several environmental factors i.e., drought, high temperature, light condition, elevated $CO_2$ level, and environmental pollution are responsible for plasticity in morphological, anatomical, and physiological traits in plants [10]. In addition, these factors could alter the physiological characteristics of plant leaves, such as lowering leaf photosynthesis and transpiration rate, stomatal conductance, and water use efficiency, and changes the pattern of growth and distribution of biomass in entire plant [11,12]. It may be vital for plant performance or survival in stressful and/or fluctuating environments, reducing ecological fitness due to anthropogenic environmental alteration. Under drought stress, longer roots improve acquisition and uptake of water from soil therefore, enabling plant species to survive in water deficit conditions [13].

Yield contributing factors in *O. europaea* include fruit number per tree [14], inflorescence and bud number, bud induction [15], inflorescence structure, flower quality, fruit set, and natural fruit drop [16]. A negative correlation exists between fruit number, biomass, and fruit oil concentration due to assimilate competition, since available assimilates are distributed and shared by more fruits [14]. Leaves are the most responsive part to climate changes [17], thus are crucial to understand adaptive strategies and plant functioning to maximize ecological fitness [18]. The ability of leaves to survive for long periods of time involves anatomical, morphological, and physiological modifications, the majority of which come at a cost and with trade-off with other processes in plants [19]. Leaf traits are more responsive to solar light [20], and it has been reported that palisade and spongy parenchyma increased with greater light intensity [21,22].

Olive (*Olea europaea* L.) is an important oil yielding, medium sized woody and evergreen tree species growing in the Mediterranean region. It has approximately 25 genera and 600 species with worldwide distribution, specifically in the temperate and tropical regions [23]. It can grow up to 8 to 15 m in height based on the prevailing climatic conditions. Fruits are initially green but later change to purple color upon maturation. These are rich sources of oil with good flavor and health benefits [24]. Pakistan is the fourth-largest oil importer country over the globe, which approximately import 70% edible oil to meet the demands [25].

*Olea europaea* is a drought tolerant species, hence has specific structural and functional modifications like reduced leaf area, leaf curling or even shedding which can significantly lower transpiration rate [26]. Leaf thickness increased under drought, mainly due to increase in palisade and spongy mesophyll. Stomata density, trichome density, epidermal cell size on abaxial and adaxial leaf surface increased under drought, while leaf area decreased [27].

In the recent decade, olive farming has gained popularity because of its high socioeconomic value. In Pakistan, olive production project was established in the Potohar region, where more than 3,166 acres of olive plants were planted [28]. According to Pakistan Agricultural Research Council (PARC), total number of trees planted in Pakistan are now 2.95 million [29]. The overall commercialization and promotion of olive cultivation and production has started in other provinces like Baluchistan, Khyber Pakhtoonkhwa and Punjab [30].

Following a review of species status, the purpose of this research was to understand how microstructural changes affected the growth and yield responses of various cultivars grown at the Barani Agricultural Research Institute (BARI), Chakwal. It was hypothesized that anatomical traits are the main contributors towards overall yield of studied cultivars. The research question to be tested in the study were: (1) which plant tissues maximally contributed to the yield enhancement? (2) to what extent anatomical and yield contributors correlate? and (3) what correlation exists between anatomical and morphological characteristics?

## Materials and methods

### Ethical approval

The study does not include any animal or human subjects and no specific ethical approval is needed. Other necessary guidelines set by University of Agriculture, Faisalabad for handling of plant material during conduction of laboratory work were followed. All samplings were done with the least possible disturbances to plant communities and environment. After completion of study, all experimental materials were properly discarded/incinerated in a controlled environment to avoid bio-contamination.

### Plant materials

Eleven cultivars were collected all over the world to correlate morpho-anatomical traits with yield contributing characteristics. The selected cultivars included two local (Pakistani) selections (BARI-2, HP Olive and QR Olive). The other cultivars were Erlik (Israel), Hamdi (Tunisia), FS-17 (Italy), Nabali (Palestine), Gemlik (Turkey), Souri (Lebanon), Manzanilla (Spain) and Azerbaijan (Azerbaiijan).

### Experimental layout and data collection

Olive cultivars were planted at Olive germplasm unit, Barani Agricultural Research Institute, Chakwal (coordinates 32˚55'36.39"N, 72˚4325.95"E, elevation 523 m a.s.l.) at an elevation of 580 m. The study site has an arid to semi-arid climate with an average annual precipitation of 786 mm during the last decade, with the maximum average summer temperatures of 36–38˚C in June, and minimum average temperatures of 1.67˚C in January. The driest months had an average rainfall of 114 mm from May to June, and a minimum of 71 mm from October to January. The plant material was collected during September to November 2017. The Olive cultivars were planted in rows with 20 m. Plants within same rows were spaced 20 m apart. The olive trees were planted at Barani Agricultural Research Institute Chakwal in the year 2010 keeping 10 m plant to plant distance. For growth and yield parameters, 6–7 years mature plants were selected and material for anatomical studies were taken from the same plant. The selected plants were uniform in shape with single stem and 3 to 4 primary branches, oriented on different directions. Briefly, six plants were randomly selected from each cultivar as replicates. The plants selected for the analysis were of uniform shape, having single stem with 3 to 4 primary branches which were oriented in different directions. Average fruit weight was determined and, after removing and cleaning the stones, flesh and stone weights were also recorded.

Each tree's youngest leaves, at the point of leaf shooting, were taken for their fresh leaf tissue. Fruit was hand-picked and weighed for each year to determine the average yield per plant [31]. The olive fruits were chosen at random from the four sides of the selected plants. With the help of weighing balance, average weight of selected fruits was calculated, and average size of fruits was measured by vernier caliper by adding fruit length and breadth and dividing on 2. After removing the fruit flesh, all of the selected stones were weighed, and the average stone weight was computed [32]. Using a portable leaf area meter (LI-3000, USA), the average leaf area was determined by selecting fully grown, healthy leaves at random from either side of the canopy. Measurements of leaf length and leaf weight were made using the method recommended by the International Olive Oil Council (IOOC).

## Anatomical studies

To investigate the anatomical traits, stems and one year old leaves were separated from 5–6 branches of upper canopy from each cultivar. Roots from 30–50 cm depth were carefully excavated at the distance of 10–20 cm from the base of the plants to obtain intact root system. Soil adhering to the roots was removed. Intact root systems and other plant parts were immediately put into polyethylene bottles and preserved in formalin acetic alcohol (FAA) solution containing (v/v) 5% formaldehyde, 50% ethanol, 10% acetic acid, and 35% distilled water for later anatomical measurements. Preserved plant material was sectioned by free-hand sectioning method, dehydrated by ethanol grades, and stained by safranin (for mechanical and vascular tissues) and fast green (for parenchymatous tissue) following [20,21]. Permanent slides were prepared by mounting in a canada balsam resin. Microstructural measurements were taken by ocular micrometer pre-calibrated with stage micro-meter. The camera-equipped stereo microscope was used to (Nikon, 104 Japan) take photographs of the sections.

## Statistical analysis

The data of various morphological, anatomical and yield parameters were subjected to analysis of variance (ANOVA) using completely randomized design with six replications. The data were using CoStat 6.4 software and means were compared by Least Significant Difference (LSD). The relationship of anatomical characteristics to yield and yield contributors was addressed by constructing clustered heatmaps (pheatmap library) between yield contributing traits (independent variables) with plant morpho-anatomical traits (response variables) in R (4.0.5). The identification of critical response of morpho-anatomical on yield and yield contributors were calculated by running a redundancy analysis (RDA) followed by constructing response curves against fruit yield (as a single factor) in Canoco (v. 4.5). For construction of RDA triplots, the cultivars were considered as fixed effect (control variable as factor 1) and yield characteristics as discrete effects (independent variables as factor 2) that influences plant morpho-anatomical attributes (response variables as factor 3). To evaluate response of individual plant attributes (morphological and root, stem, and leaf anatomical) with fruit yield, response curves were drawn by fitting a Generalized Linear Model (GLM) in Canoco.

## Results

### Morphological and yield traits

The height of the plants varied significantly depending on the cultivar, reaching a maximum in three cultivars (Erlik, Hamdi, and BARI-2), and minimum in HP Olive and QR Olive. All cultivars displayed a substantial variance in trunk circumference. The QR Olive and HP Olive had the most leaves per branch, whereas the BARI-2 had the fewest (Table 1). BARI-2 had the

**Table 1. Morphological and yield traits of *Olea europaea* L. cultivars being grown at Barani Agricultural Research Institute, Chakwal, Pakistan (*n* = 6).**

| Characteristics | Erlik | Hamdi | BARI-2 | FS-17 | Nabali | Gemlik | Souri | Manzanilla | Azerbaiijan | HP Olive | QR Olive | F-ratio |
|---|---|---|---|---|---|---|---|---|---|---|---|---|
| PH (m) | 5.5a | 5.5a | 5.3ab | 5.0abc | 5.0abc | 5.0abc | 4.9bcd | 4.9bcd | 4.9bcd | 4.4d | 4.6cd | 8.2*** |
| TC (m) | 0.9a | 0.8ab | 0.8ab | 0.9a | 0.8ab | 0.9a | 0.8bc | 0.8bc | 0.7cd | 0.7d | 0.7d | 20.9*** |
| NL | 180cd | 185bc | 153f | 175de | 170e | 185bc | 190b | 187bc | 193b | 205a | 210a | 46.5*** |
| NF | 27b | 25bcd | 30a | 24cd | 20e | 23d | 20e | 18e | 19e | 26bc | 27b | 30.7*** |
| LL (cm) | 7bc | 7.1b | 7.4a | 6.8c | 6.9bc | 6.3d | 5.8e | 5.9e | 5.8e | 6e | 4.9f | 168.8*** |
| LW (cm) | 1.4cd | 1.5bc | 1.7b | 1.6bc | 1.9a | 1.2de | 1.1c | 1.2de | 1.2de | 1.1e | 1.1e | 23.0*** |
| LA (cm$^2$) | 7.4c | 8.0b | 9.4a | 8.2b | 9.8a | 5.7d | 4.8f | 5.3de | 5.2ef | 5.0ef | 3.4g | 36.5*** |
| FL (cm) | 2.5bc | 2.9a | 2.4cd | 2.2d | 2.6b | 2.3cd | 2.3cd | 2.4cd | 2.0e | 1.3f | 1.4f | 82.6*** |
| FD (cm) | 1.7abc | 2.1a | 1.5cd | 1.7abc | 2ab | 1.6bcd | 1.6bcd | 1.6bcd | 1.7abc | 1.2d | 1.2d | 7.5*** |
| FW (g plant$^{-1}$) | 4.5bc | 5.2a | 3.4e | 4.8bc | 4.9ab | 4.5bc | 4.4c | 4.5bc | 4.0d | 1.0f | 1.0f | 189.0*** |
| SW (g plant$^{-1}$) | 0.7a | 0.7a | 0.5cd | 0.6b | 0.4d | 0.4d | 0.6b | 0.5bc | 0.5cd | 0.3e | 0.3e | 78.8*** |
| SL (cm) | 1.6abc | 1.7a | 1.3cd | 1.4bcd | 1.3cd | 1.6cb | 1.5bcd | 1.5bcd | 1.3d | 1.2d | 1.3cd | 7.2*** |
| SD (cm) | 0.8ab | 0.8ab | 0.8abc | 0.8ab | 0.6cd | 0.7bcd | 0.8ab | 0.8a | 0.6cd | 0.5e | 0.6de | 10.2*** |
| FY (g) | 27.1a | 20.3ab | 19.1ab | 18.2ab | 17.1ab | 17.1ab | 14.6bc | 12.9bc | 11.1bc | 4.2c | 4.2c | 6.7*** |

Means followed by the same letters within rows are not significantly different (P≤0.05).

**Morphology:** PH: Plant height, TC: Trunk circumference, NL: Number of leaves per branch, NF: Number of fruits per branch (Lowest branch), LL: Leaf length, LW: Leaf width, LA: Leaf area, FL: Fruit length, FD: Fruit diameter, FW: Fruit weight, SW: Stone weight, SL: Stone length, SD: Stone diameter, FY: Fruit yield per branch.

most fruits per branch, compared to Manzanilla and Azerbaijan cultivars, which had the fewest. BARI-2 showed the maximum leaf length and QR Olive showed the minimum leaf length. Leaf width was the maximum in Nabali, while the minimum in Gemlik, Souri, Manzanilla, Azerbaijan, HP Olive and QR Olive. The maximum leaf area was observed in the Nabali and the minimum in the QR Olive (Table 1). Fruit length, fruit diameter and fruit weight were the maximum in the Hamdi, while the minimum HP Olive and QR Olive (Table 1). Stone weight was the maximum in the Hamdi, while the minimum in HP Olive (Table 1). Stone diameter was maximum in Erlik, Hamdi, BARI-2 and FS-17. Fruit yield was the maximum in the Erlik, while the minimum in the HP Olive and QR Olive (Table 1).

## Anatomical parameters

**Root anatomical parameters.** Root cross-sectional area did not show any significant difference and remained similar in all cultivars. The maximum epidermal thickness was recorded in the Erlik, while the minimum was observed in the Hamdi, BARI-2, FS-17, Nabali and Gemlik. QR Olive showed the thickest cortical region and cortical cell area, while the Erlik showed the thinnest cortical region and cortical cell area (Table 2). Sclerenchymatous thickness was the highest in BARI-2 and the lowest in QR Olive. The maximum collenchymatous thickness was recorded in the QR Olive and the minimum was observed in the Erlik and Hamdi cultivars (Table 2). Phloem thickness was the maximum in the Erlik and the minimum in the QR Olive. Metaxylem area was the maximum in Nabali, while the minimum in the QR Olive. In all cultivars, the pith cross-sectional area showed no significant difference (Table 2). QR Olive showed the largest, while the Hamdi showed the smallest pith cell area (Table 2, Fig 1).

**Stem anatomical parameters.** All the olive cultivars showed no significant difference in stem cross sectional area and in sclerenchymatous thickness (Table 2, Fig 2). Epidermal thickness, cortical region thickness and cortical cell area was the maximum in HP Olive and QR Olive, while the minimum was observed in Erlik and Hamdi (Table 2). The maximum sclerenchymatous thickness was observed in the FS-17, while the minimum was recorded in the Manzanilla and Azerbaijan. Collenchymatous thickness, phloem thickness and metaxylem

**Table 2. Anatomical traits of *Olea europaea* L. culltivers being grown at Barani Agricultural Research Institute, Chakwal, Pakistan (*n* = 6).**

| Characteristic | Erlik | Hamdi | BARI-2 | FS-17 | Nabali | Gemlik | Souri | Manzanilla | Azerbaiijan | HP Olive | QR Olive | F-ratio |
|---|---|---|---|---|---|---|---|---|---|---|---|---|
| **Root anatomy** | | | | | | | | | | | | |
| RCS (mm$^2$) | 0.72 | 0.70 | 0.70 | 0.72 | 0.69 | 0.71 | 0.71 | 0.72 | 0.72 | 0.73 | 0.73 | 1.3$^{NS}$ |
| RET (μm) | 19.5a | 13.2c | 13.2c | 13.9c | 13.2c | 13.9c | 15.9b | 15.9b | 15.9b | 16.3b | 16.6b | 6.62*** |
| RCT (μm) | 44.5e | 46.6de | 47.3de | 49.4cd | 49.4cd | 50.8bc | 51.5bc | 51.5bc | 52.9ab | 53.6ab | 54.5a | 16.2*** |
| RCA (μm$^2$) | 11.9d | 9.5d | 11.2d | 12.8cd | 12.8cd | 17.6bc | 17.6bc | 18.6b | 20.8ab | 22.0ab | 24.4a | 16.3*** |
| RST (μm) | 36.1ab | 36.1ab | 38.9a | 35.9ab | 35.9ab | 35.6ab | 32.8bc | 32.0bc | 30.6bc | 30.4bc | 29.9c | 5.9*** |
| ROT (μm) | 63.9e | 63.9e | 65.3de | 66.4cde | 66.3cde | 68.1bcde | 69.5bcd | 70.6bc | 72.3b | 75.8a | 77.8a | 18.5*** |
| RPT (μm) | 43.1a | 39.6b | 38.9bc | 38.9bc | 36.1bcd | 36.1bcd | 34.8cd | 34.8cd | 32.7de | 32.0de | 29.9e | 15.3*** |
| RMA (μm$^2$) | 105.0bcd | 108.5bc | 108.8bc | 115.6b | 131.9a | 98.6cde | 96.0def | 94.8def | 87.5ef | 86.5ef | 84.9f | 22.6*** |
| RPA (μm$^2$) | 11.9e | 9.5e | 11.2f | 12.8de | 12.8cde | 17.6cd | 17.6bc | 18.6ab | 20.8a | 22.0a | 24.4a | 26.3*** |
| **Stem anatomy** | | | | | | | | | | | | |
| SCS (mm$^2$) | 0.80 | 0.80 | 0.81 | 0.82 | 0.81 | 0.81 | 0.83 | 0.82 | 0.82 | 0.82 | 0.82 | 0.9$^{NS}$ |
| SET (μm) | 4.5d | 4.5d | 4.7cd | 5.1bcd | 5.1bcd | 5.4abcd | 5.8abcd | 6.4abc | 6.7ab | 7.1a | 7.1a | 6.0*** |
| SCT (μm) | 44.5e | 44.5e | 46.6de | 47.3cde | 48.7bcde | 48.7bcde | 50.7abcd | 51.4abc | 52.8ab | 52.8ab | 53.5a | 10.7*** |
| SCA (μm$^2$) | 14.6d | 15.6cd | 18.6cd | 17.6cd | 18.6cd | 19.8c | 25.6b | 24.3b | 26.8b | 30.7a | 33.6a | 31.5*** |
| SST (μm) | 26.4 | 23.6 | 23.6 | 27.8 | 26.4 | 26.4 | 26.4 | 22.2 | 22.2 | 26.4 | 26.4 | 2.0$^{NS}$ |
| SOT (μm) | 45.2a | 44.5ab | 43.1abc | 43.1abc | 42.4abc | 40.3bcd | 40.3bcd | 39.6cd | 38.9cd | 37.5de | 34.8e | 9.3*** |
| SPT (μm) | 54.9a | 54.9a | 53.5a | 53.5a | 50.7b | 49.4bc | 48.7bc | 48.0bc | 46.6cd | 44.8d | 44.5d | 26.7*** |
| SMA (μm$^2$) | 43.2a | 41.7ab | 40.9ab | 40.9ab | 38.6abc | 37.7bc | 35.5cd | 34.1cd | 33.9cd | 35.5cd | 32.0d | 9.0*** |
| SPA (μm$^2$) | 125.7f | 152.6e | 148.9e | 170.6de | 177.2d | 187.1cd | 204.3bc | 207.9bc | 222.3ab | 237.2a | 233.6a | 7.4*** |
| **Leaf anatomy** | | | | | | | | | | | | |
| LMT (μm) | 419.8a | 418.4a | 414.2b | 408.0b | 409.4cd | 405.2bc | 405.2d | 401.0d | 401.0e | 397.5e | 408.0f | 59.8*** |
| LLT (μm) | 196.7ab | 198.4a | 195.3ab | 194.3ab | 191.0bc | 184.5cd | 186.0cd | 186.0cd | 186.0cd | 183.8d | 188.5cd | 11.7*** |
| LST (μm) | 111.2bc | 111.9bc | 113.3bc | 113.3b | 107.7d | 107.7d | 109.1cd | 111.9bc | 116.1a | 111.9bc | 116.1a | 16.9*** |
| LPT (μm) | 73.7a | 73.7a | 68.1b | 67.6b | 68.8b | 63.3c | 61.2cd | 59.1de | 57.0e | 56.9e | 56.9e | 42.6*** |
| LUT (μm) | 7.1c | 7.1c | 7.2c | 7.7bc | 7.7bc | 7.6bc | 8.3abc | 8.2abc | 8.5ab | 9.2a | 8.9a | 9.0*** |
| LDT (μm) | 6.9 | 6.9 | 7.7 | 7.7 | 7.9 | 7.9 | 7.0 | 9.0 | 8.1 | 8.3 | 7.9 | 1.6$^{NS}$ |
| LBT (μm) | 4.9 | 5.9 | 6.3 | 5.9 | 6.6 | 6.6 | 6.7 | 6.0 | 7.0 | 6.5 | 7.5 | 1.3$^{NS}$ |
| LMA (μm$^2$) | 20.9ab | 21.9a | 18.7ab | 18.5ab | 19.5ab | 18.1ab | 17.7ab | 18.5ab | 16.8ab | 17.3ab | 15.2b | 2.4* |
| LHT (μm) | 77.8a | 75.8a | 76.9a | 72.7b | 71.6b | 69.5b | 70.2b | 66.0c | 63.3d | 61.0d | 60.3d | 46.3*** |
| LCT (μm) | 111.2b | 111.2b | 111.9b | 111.4b | 111.2b | 115.4b | 119.6a | 120.3a | 119.5a | 122.3a | 122.3a | 23.1*** |
| LCA (μm$^2$) | 3.9c | 3.9c | 5.0bc | 6.1abd | 5.5abc | 6.8abc | 8.1abc | 8.1abc | 8.1abc | 9.6a | 8.8ab | 5.0*** |

Means followed by the same letters within rows are not significantly different (P≤0.05).

**Root anatomy:** RCS-Root cross sectional area, RET-Epidermal thickness, RCT-Cortical region thickness, RCA-Cortical cell area, RST-Sclerenchymatous thickness, ROT-Collenchymatous thickness, RPT-Phloem thickness, RMA-Metaxylem vessel diameter, RPA-Pith cell area. **Stem anatomy:** SCS-Stem cross-sectional area, SET-Epidermal thickness, SCT-Cortical region thickness, SCA-Cortical cell area, SST-Sclerenchymatous thickness, SOT-Collenchymatous thickness, SPT-Phloem thickness, SMA-Metaxylem vessel diameter, SPA-Pith cell area. **Leaf anatomy:** LMT-Midrib thickness, LLT-Lamina thickness, LST-Spongy cell thickness, LPT-Palisade cell thickness, LUT-Cuticle thickness, LDT-Adaxial epidermal thickness, LBT-Abaxial epidermal thickness, LMA-Metaxylem vessel diameter, LHT-Phloem thickness, LCT-Parenchymatous region thickness, LCA-Parenchymatous cell area.

thickness were the maximum in the Erlik, while the lowest in the QR Olive. Phloem thickness was also maximum in the Hamdi and the minimum in the HP Olive. The pith cell area was the maximum in HP Olive and was the minimum in the Erlik (Table 2, Fig 2).

**Leaf anatomical parameters.** The maximum midrib and lamina thickness was recorded in the Erlik, while the minimum in the HP Olive. Spongy cell thickness was the maximum in the Azerbaiijan and QR Olive, while the minimum in the Nabali and Gemlik (Table 2, Fig 3). Palisade cell thickness was the maximum in the Erlik and Hamdi, while the minimum in the

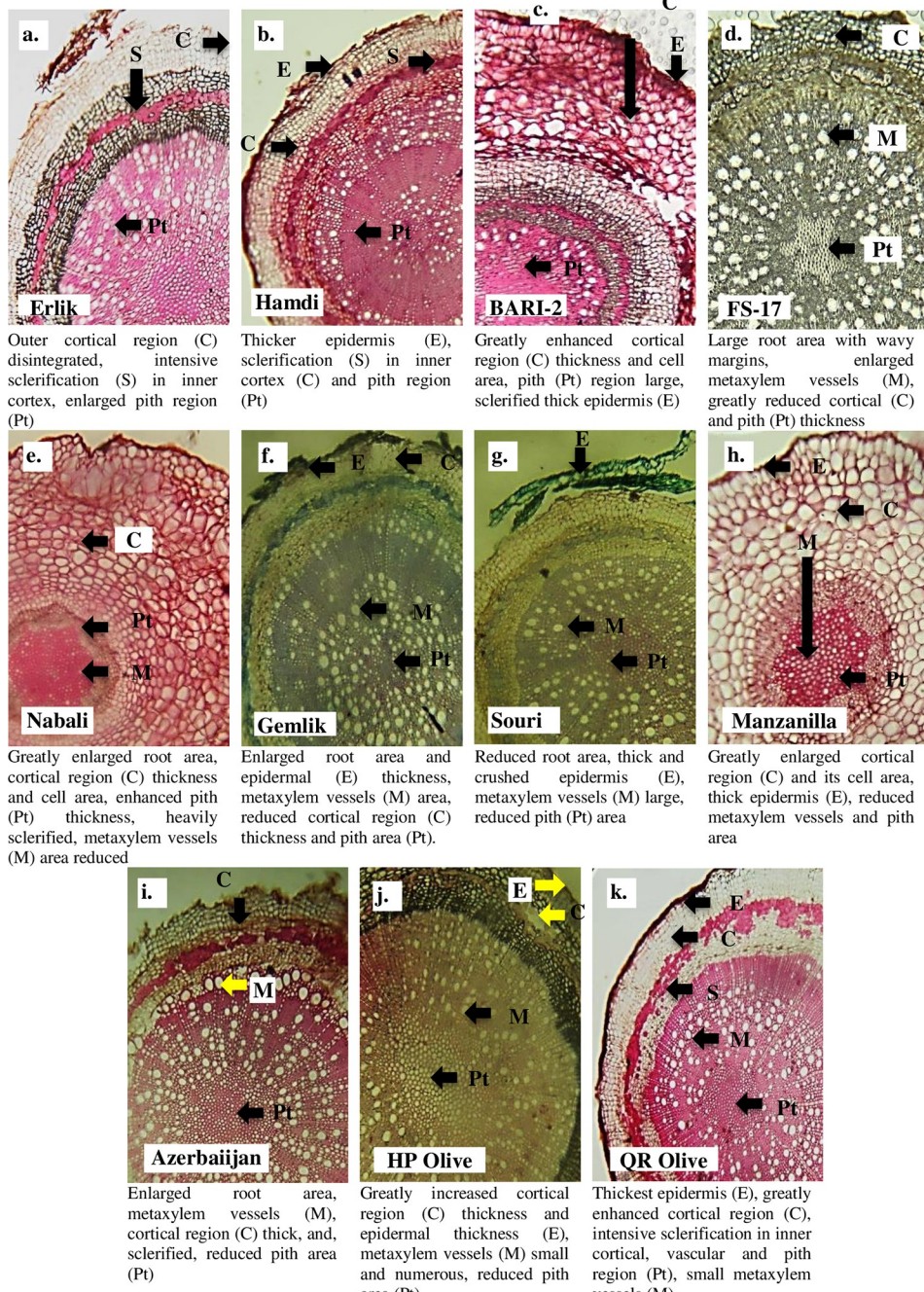

**Fig 1. Root transverse sections of *Olea europaea* L. cultivars planted at Barani Agricultural Research Institute, Chakwal.**

HP Olive and QR Olive. Cuticle thickness was the maximum in HP Olive, while the minimum in the Erlik and Hamdi (Table 2). The maximum adaxial epidermal thickness was recorded in the Manzanilla, while the minimum in the Erlik and Hamdi. Abaxial epidermal thickness was the maximum QR Olive, while the minimum in Erlik and Hamdi (Table 2). The maximum metaxylem area was observed in the Hamdi, while the minimum was recorded in the QR Olive. Phloem thickness was the maximum in the Erlik and the minimum in the QR Olive

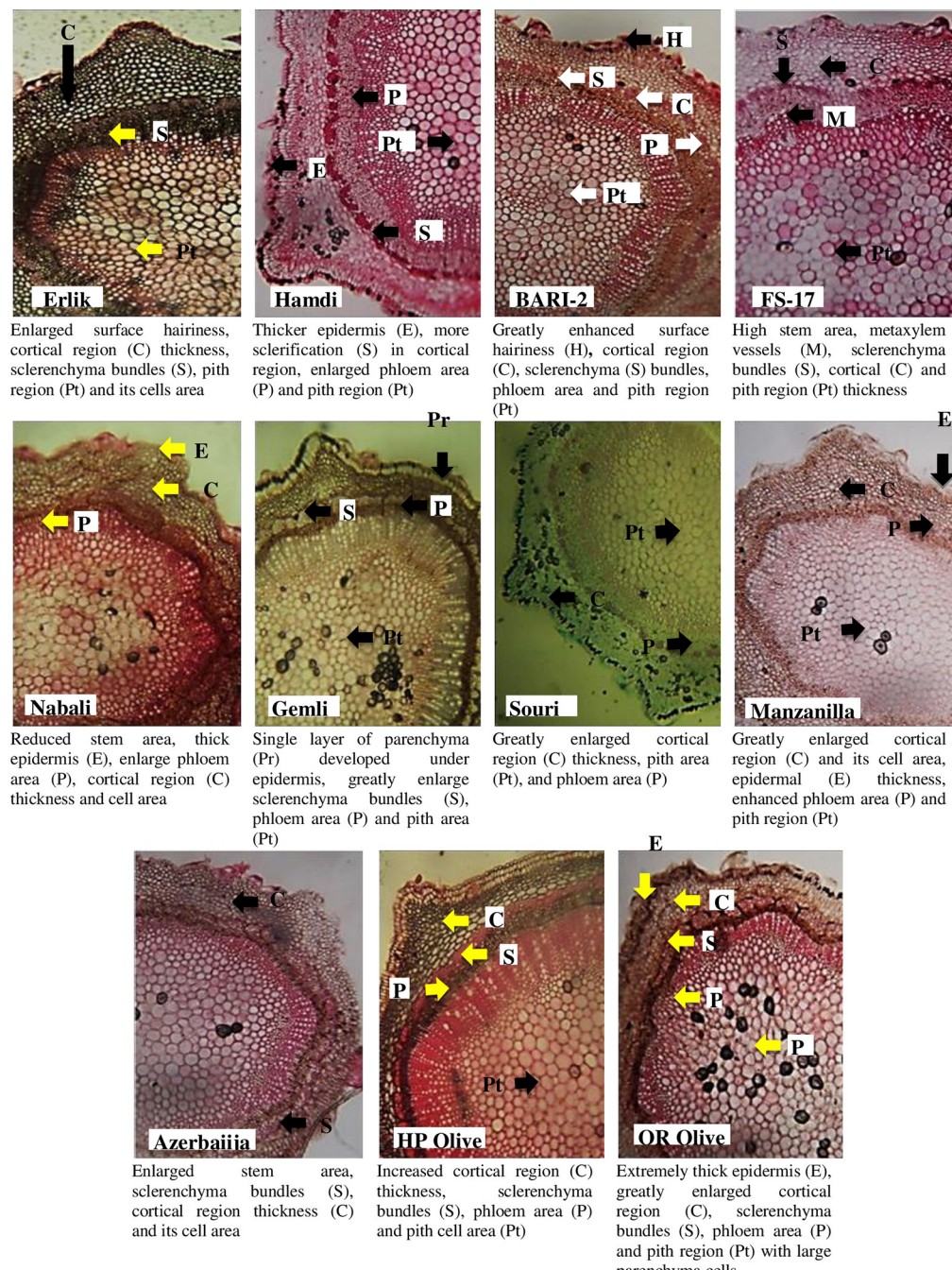

**Fig 2. Stem transverse sections of *Olea europaea* L. cultivars planted at Barani Agricultural Research Institute, Chakwal.**

(Table 2). The maximum cortical region thickness was recorded in the HP Olive and QR Olive, while almost all other cultivars showed the same cortical region thickness. Cortical cell area was the maximum in the HP Olive, while the minimum in the Erlik and Hamdi (Table 2, Fig 3).

**Redundancy analysis.** Yield attributes including fruit diameter, fruit weight, fruit length, fruit yield, seed weight, seed length and seed diameter were closely grouped with leaf area in

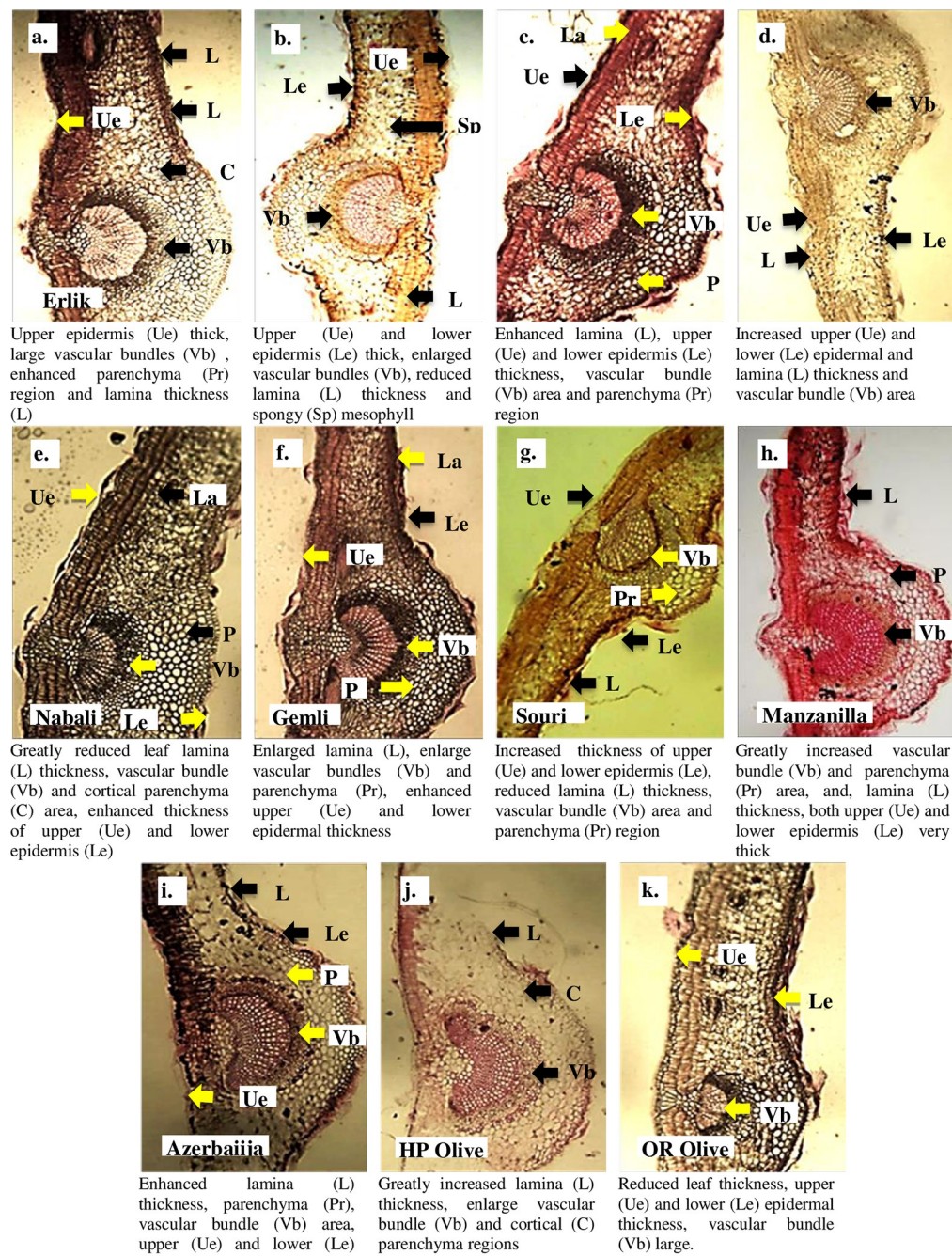

**Fig 3. Leaf transverse sections of *Olea europaea* L. cultivars planted at Barani Agricultural Research Institute, Chakwal.**

Nabali, Hamdi and FS-17 cultivars. All other growth parameters did not showed grouping with yield parameters but some growth attributes such as leaf length, number of leaves per plant, plant height, leaf width, and trunk circumference were closely grouped with each other in Azerbaiijan, Manzanilla, Gemlik and Souri cultivars (Fig 4A).

Fruit diameter, fruit weight, fruit length, fruit yield, seed weight, seed length, and seed diameter were closely grouped with root anatomical traits such as root pith cell area,

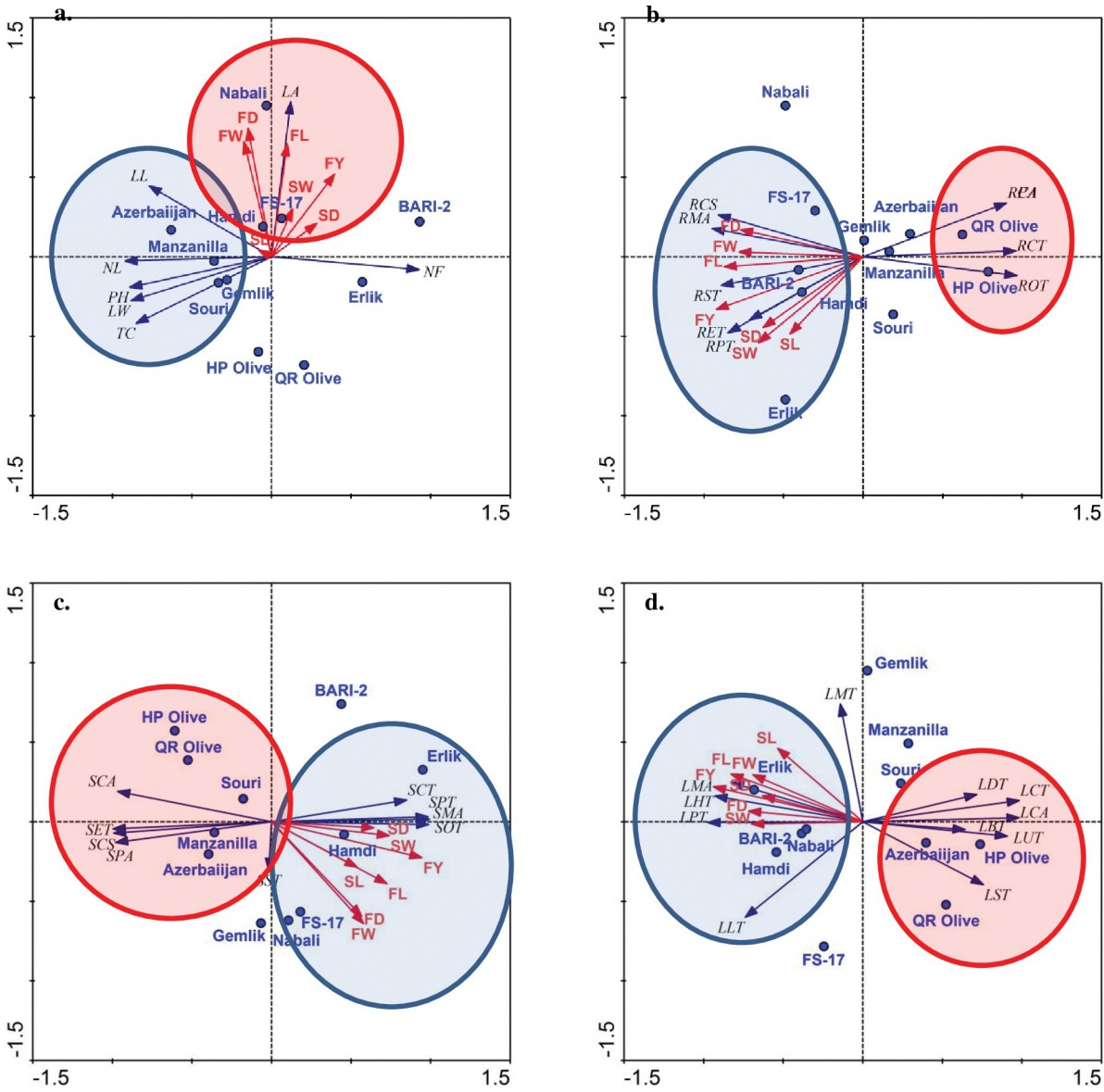

**Fig 4.** Relationship of yield parameters with a) growth, b) root anatomical, c) stem anatomical and d) leaf anatomical traits of *Olea europaea cultivars* planted at Barani Agricultural Research Institute, Chakwal. **Morphology:** PH: Plant height, TC: Trunk circumference, NL: Number of leaves per branch, NF: Number of fruits per branch (Lowest branch), LL: Leaf length, LW: Leaf width, LA: Leaf area, FL: Fruit length, FD: Fruit diameter, FW: Fruit weight, SW: Stone weight, SL: Stone length, SD: Stone diameter, FY: Fruit yield. **Root anatomy:** RCS-Root cross sectional area, RET-Epidermal thickness, RCT-Cortical region thickness, RCA-Cortical cell area, RST-Sclerenchymatous thickness, ROT-Collenchymatous thickness, RPT-Phloem thickness, RMA-Metaxylem vessel diameter, RPA-Pith cell area. **Stem anatomy:** SCS-Stem cross-sectional area, SET-Epidermal thickness, SCT-Cortical region thickness, SCA-Cortical cell area, SST-Sclerenchymatous thickness, SOT-Collenchymatous thickness, SPT-Phloem thickness, SMA-Metaxylem vessel diameter, SPA-Pith cell area. **Leaf anatomy:** LMT-Midrib thickness, LLT-Lamina thickness, LST-Spongy cell thickness, LPT-Palisade cell thickness, LUT-Cuticle thickness, LDT-Adaxial epidermal thickness, LBT-Abaxial epidermal thickness, LMA-Metaxylem vessel diameter, LHT-Phloem thickness, LCT-Parenchymatous region thickness, LCA-Parenchymatous cell area.

metaxylem area, sclerenchymatous thickness, epidermal thickness and phloem thickness in FS-17, BARI-2, Erlik and Hamdi cultivars. Other root anatomical traits like cortical cell area, cortical region thickness and collenchyma thickness were closely grouped with each other in HP Olive and QR Olive cultivars (Fig 4B).

Yield contributing parameters like fruit diameter, weight length, and yield, and seed weight, length, and diameter were closely grouped with stem anatomical traits, i.e., stem cortical region thickness, phloem thickness, metaxylem thickness and collenchymatous thickness in FS-17, Nabali, Erlik and Hamdi cultivars. Other stem anatomical traits, i.e., cortical cell area, epidermal thickness, cross sectional area and phloem area were closely grouped with each other in Azerbaiijan, Manzanilla, Souri, HP Olive and QR Olive cultivars (Fig 4C).

Fruit seed traits like weight, length, and diameter along with fruit yield, were closely grouped with leaf anatomical traits, i.e., metaxylem area, phloem thickness, palisade thickness and lamina thickness in Erlik, BARI-2, Nabali, and Hamdi cultivars. Other leaf anatomical traits like adaxial epidermal thickness, parenchymatous thickness, cortical cell area, abaxial epidermal thickness, cuticle thickness, and spongy thickness were closely grouped with each other in Azerbaiijan, HP Olive and QR Olive cultivars (Fig 4D).

**Response curve.** Growth parameters, i.e., plant height, trunk circumference, leaf length, leaf area, leaf width, and number of fruits per branch showed positive response, while number of leaves per plant showed negative response with increase in fruit yield (Fig 5A). Root anatomical traits, i.e., pith cell area, collenchymatous thickness, cortical region thickness, cortical cell area and pith area showed negative response, while sclerenchymatous thickness, phloem thickness and metaxylem area showed positive response along with increase in fruit yield. Root epidermal thickness showed linear response with fruit yield (Fig 5B). Stem anatomical traits like phloem thickness, cortical region thickness, epidermal thickness, cross sectional area and cortical cell area showed negative slope, while phloem thickness, collenchyma thickness and metaxylem thickness showed positive slope with fruit yield. Stem sclerenchymatous thickness showed linear response with fruit yield (Fig 5C). Leaf anatomical traits, i.e., spongy thickness, parenchymatous thickness, cuticle thickness, cortical cell area and abaxial epidermal thickness showed negative slope, while phloem thickness, lamina thickness, midrib thickness, palisade thickness and metaxylem area showed positive slope with fruit yield (Fig 5D).

**Clustered heatmap.** Clustered heatmap were constructed to understand the contribution of each trait of individual cultivar towards yield. The heatmap of yield and growth attributes showed grouping of HP Olive and QR Olive (cluster 1), Erlik and Hamdi (cluster 2), Azerbaiijan, Gemlik, Souri and Manzanilla (cluster 3), and Nabali, BARI-2 and FS-17 (cluster 4). Fruit yield, seed clustered with trunk circumference, plant height and shoot length in cultivars HP Olive and QR Olive, while strong positive grouping Erlik and Hamdi. A close association of fruit diameter, fruit length and fruit weight were observed, which clustered negatively in HP Olive and Q. Number of leaves strongly positively associated in HP Olive and QR Olive and strongly negatively in Nabali, BARI-2 and FS-17 (Fig 6A).

The heatmap among yield and root anatomical traits showed three distinct assemblages of olive cultivars. HP Olive and QR Olive clustered together, while Nabali, BARI-2 and FS-17 clustered in a separate group and the rest in third group. Seed diameter, seed length and seed weight strongly and negatively clustered in HP Olive and QR Olive, while strongly positively in Erlik and Hamdi. Fruit traits (yield, length, weigh and diameter) clustered with root sclerenchymatous thickness, phloem thickness and metaxylem vessel diameter. A strong negative correlation was noted in HP Olive and QR Olive and strong positive was seen in Erlik and Hamdi. Cortical cell area, pith cell area, cortical region thickness and collenchymatous showed close grouping, here a strong negative association was seed in Erlik and Hamdi and strong positive in HP Olive and QR Olive. Epidermal thickness closely clustered with root cross-sectional area, where strong positive correlation was recorded in HP Olive, QR Olive and Erlik and strong negative in Hamdi, Nabali and BARI-2 (Fig 6B).

The relationship among yield and stem anatomical traits showed two distinct clustered, the first of HP Olive, QR Olive, Souri, Manzanilla, and Azerbaijan and the second of Erlik, Hamdi,

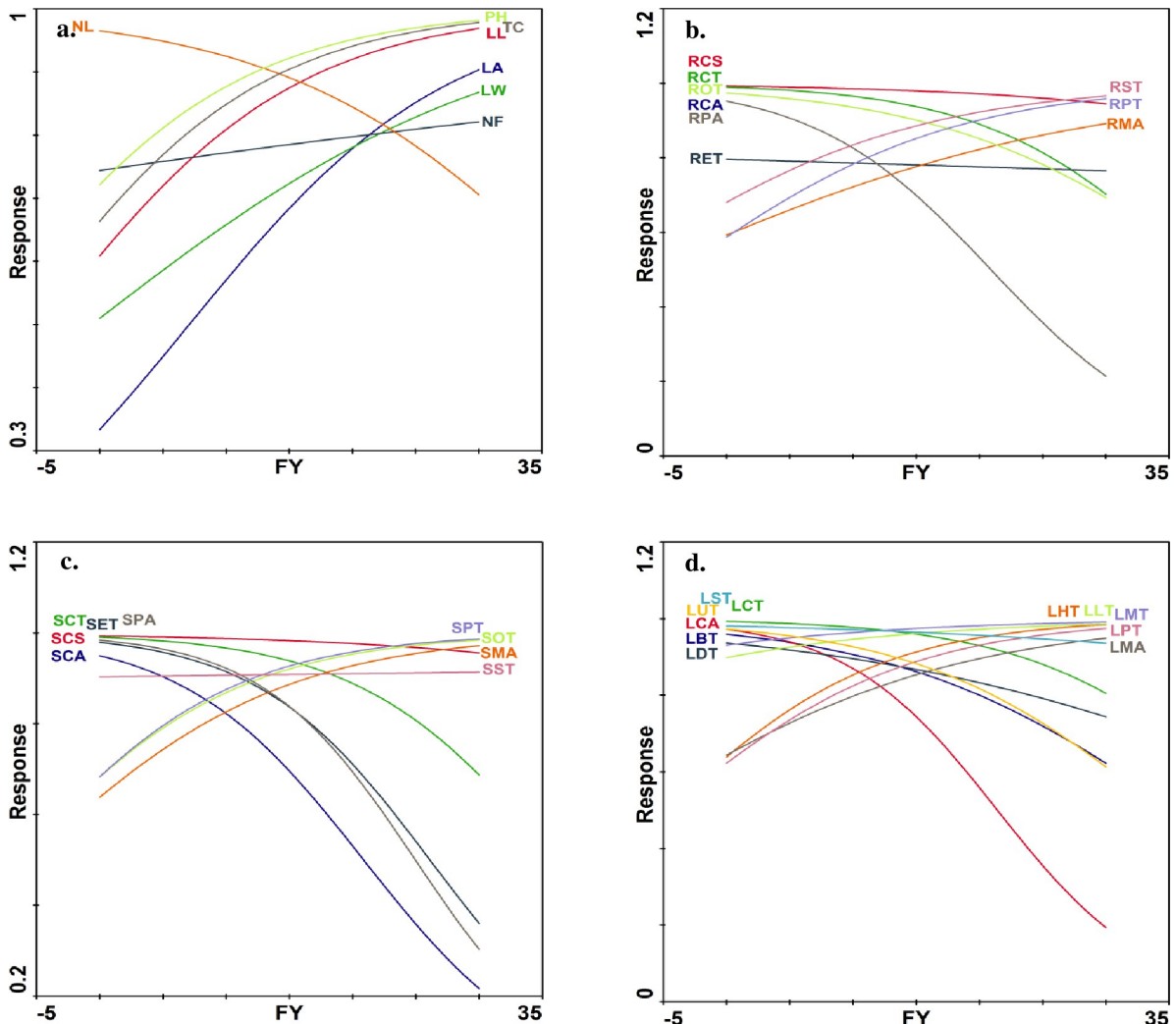

**Fig 5.** Response of a) growth, b) root anatomical, c) stem anatomical and d) leaf anatomical traits with fruit yield of *Olea europaea cultivars* planted at Barani Agricultural Research Institute, Chakwal. **Morphology:** PH: Plant height, TC: Trunk circumference, NL: Number of leaves per branch, NF: Number of fruits per branch (Lowest branch), LL: Leaf length, LW: Leaf width, LA: Leaf area, FL: Fruit length, FD: Fruit diameter, FW: Fruit weight, SW: Stone weight, SL: Stone length, SD: Stone diameter, FY: Fruit yield. **Root anatomy:** RCS-Root cross sectional area, RET-Epidermal thickness, RCT-Cortical region thickness, RCA-Cortical cell area, RST-Sclerenchymatous thickness, ROT-Collenchymatous thickness, RPT-Phloem thickness, RMA-Metaxylem vessel diameter, RPA-Pith cell area. **Stem anatomy:** SCS-Stem cross-sectional area, SET-Epidermal thickness, SCT-Cortical region thickness, SCA-Cortical cell area, SST-Sclerenchymatous thickness, SOT-Collenchymatous thickness, SPT-Phloem thickness, SMA-Metaxylem vessel diameter, SPA-Pith cell area. **Leaf anatomy:** LMT-Midrib thickness, LLT-Lamina thickness, LST-Spongy cell thickness, LPT-Palisade cell thickness, LUT-Cuticle thickness, LDT-Adaxial epidermal thickness, LBT-Abaxial epidermal thickness, LMA-Metaxylem vessel diameter, LHT-Phloem thickness, LCT-Parenchymatous region thickness, LCA-Parenchymatous cell area.

Nabali, BARI-2 and FS)17. Two sub-clusters in each major cluster were recorded, where Erlik and Hamdi, and HP Olive and QR Olive closely clustered with each other. Fruit yield closely clustered with stem metaxylem diameter, phloem thickness and collenchymatous thickness. A strong positive correlation was recorded in Erlik, Hamdi, BARI-2 and FS-17, while a strong negative in HP Olive, QR Olive, Souri, Manzanilla and Azerbaiijan. Stem sclerenchymatous thickness responded independently, where a strong positive correlation was observed in FS-17, Erlik, Nabali, Gemlik, HP Olive, QR Olive and Sour, and a strong negative in BARI-2, Manzanilla and Azerbaiijan. Stem cross-sectional area, cortical cell area, epidermal thickness,

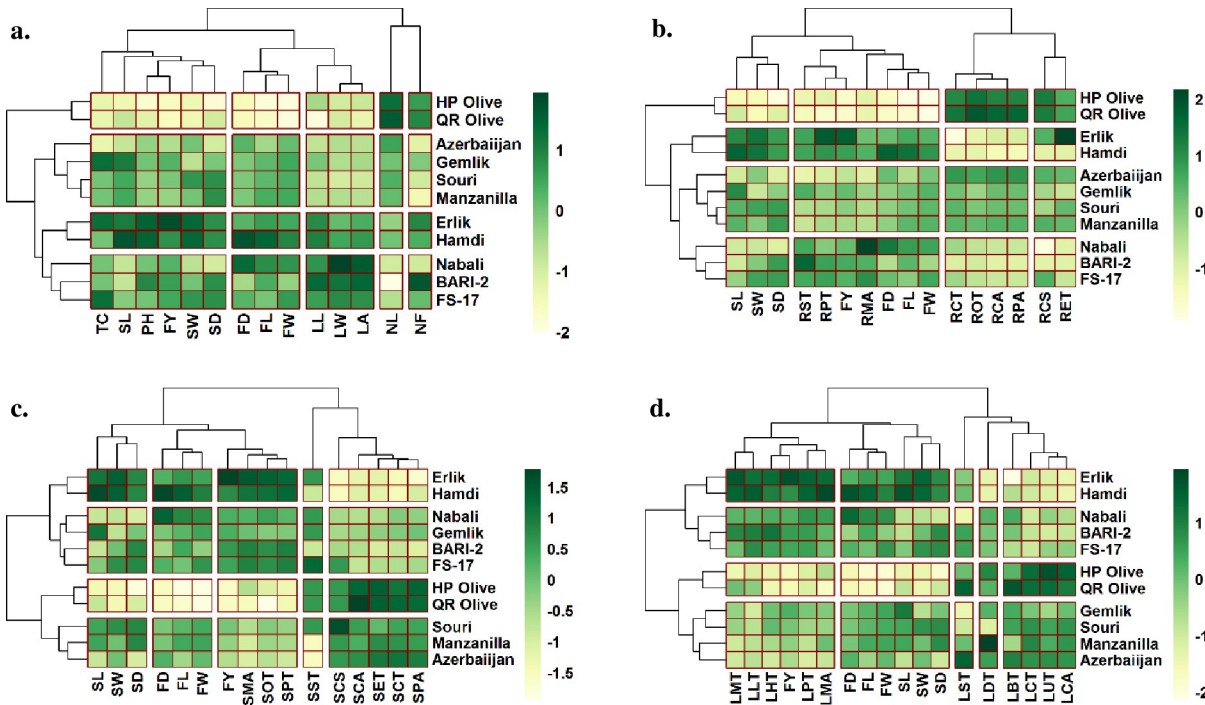

**Fig 6.** Heatmaps showing clustering of yield parameter with a) growth, b) root anatomical, c) stem anatomical and d) leaf anatomical traits of *Olea europaea* cultivars planted at Barani Agricultural Research Institute, Chakwal. **Morphology:** PH: Plant height, TC: Trunk circumference, NL: Number of leaves per branch, NF: Number of fruits per branch (Lowest branch), LL: Leaf length, LW: Leaf width, LA: Leaf area, FL: Fruit length, FD: Fruit diameter, FW: Fruit weight, SW: Stone weight, SL: Stone length, SD: Stone diameter, FY: Fruit yield. **Root anatomy:** RCS-Root cross sectional area, RET-Epidermal thickness, RCT-Cortical region thickness, RCA-Cortical cell area, RST-Sclerenchymatous thickness, ROT-Collenchymatous thickness, RPT-Phloem thickness, RMA-Metaxylem vessel diameter, RPA-Pith cell area. **Stem anatomy:** SCS-Stem cross-sectional area, SET-Epidermal thickness, SCT-Cortical region thickness, SCA-Cortical cell area, SST-Sclerenchymatous thickness, SOT-Collenchymatous thickness, SPT-Phloem thickness, SMA-Metaxylem vessel diameter, SPA-Pith cell area. **Leaf anatomy:** LMT-Midrib thickness, LLT-Lamina thickness, LST-Spongy cell thickness, LPT-Palisade cell thickness, LUT-Cuticle thickness, LDT-Adaxial epidermal thickness, LBT-Abaxial epidermal thickness, LMA-Metaxylem vessel diameter, LHT-Phloem thickness, LCT-Parenchymatous region thickness, LCA-Parenchymatous cell area.

cortical region thickness and pith cell area closely assembled in a separate cluster. A strong positive correlation was noted in HP Olive, QR Olive, Souri, Manzanilla and Azerbaiijan, and a strong negative in Erlik, Hamdi, BARI-2 and FS-17 (Fig 6C).

The clustered heatmap of yield and leaf anatomical traits showed two major clusters, the first one of Erlik, Hamdi, Nabali, BARI-2 and FS-17l, and the second of HP Olive, QR Olive, Gemlik, Souri, Manzanilla and Azerbaiijan. All yield contributors were clustered in a separate group, where a strong positive association was recorded in Erlik and Hamdi, and a strong negative in HP Olive and QR Olive. Midrib thickness, lamina thickness, phloem thickness, palisade cell thickness and metaxylem vessel diameter closely clustered with fruit yield. A strong positive correlation was seen in Erlik and Hamdi, while a strong negative in HP Olive and QR Olive. Abaxial epidermal thickness, parenchymatous region thickness, parenchymatous cell area and cuticle thickness grouped in a close cluster, where a strong positive correlation was recorded in HP Olive and QR Olive, and a strong negative in Erlik, Hamdi, Nabali, BARI-2 and FS-17. Spongy cell thickness and adaxial epidermal thickness showed a independent behaviour and not clustered with any other yield or leaf anatomical trait (Fig 6D).

**Pearson's correlation coefficients.** Plant height and trunk circumference were positively correlated with all yield-contributing traits with few exceptions (Table 3). Fruit length and total yield showed stronger correlations (p>0.001). A positive correlation of leaf length was

noted with fruit length (p>0.05), fruit diameter and yield (p>0.01). Root anatomical characteristics were generally negatively correlated with yield-contributing traits, where fruit yield exhibited a strong negative correlation (p>0.001) with cortical thickness and its cell area, collenchymatous thickness and pith cell area. A strong negative correlation (p>0.001) of fruit length was calculated with cortical cell area and collenchymatous thickness (Table 3). Root phloem thickness was positively correlated with fruit yield (p>0.001), fruit length (p>0.01) and stone weight (p>0.01).

Fruit yield was strong negatively correlated (p>0.001) with stem anatomical characteristics like epidermal thickness, cortical thickness and cell area, and pith cell area. A significant positive correlation of fruit yield was recorded with collenchymatous thickness, phloem thickness and metaxylem diameter (Table 3). Stem epidermal thickness, cortical thickness and cortical cell area were negatively correlated with most of the yield-contributing traits, while collenchymatous thickness and phloem thickness positively correlated with yield traits. Fruit yield was negatively correlated (p>0.001) with leaf traits like cuticle thickness, parenchymatous thickness, parenchyma cells, whereas was positively correlated (p>0.001) with palisade thickness and phloem thickness (Table 3). Leaf cuticle thickness was generally negatively correlated while metaxylem diameter and phloem thickness was positively correlated with yield traits.

Fig 7 revealed a significant relationship between root, stem and leaf anatomical characteristics. Root anatomical attributes RCS, RCA, RPA, ROP and RCT were positively correlated with stem SCS, SET, SCA and SPA, while they were all negatively related to SPT, SMA, and SOT (Fig 7A). A correlation matrix between root and leaf anatomical traits revealed a significant positive correlation between root RST, RMA, RPT and leaf LPT, LHT, LMT, LLT. All of the aforementioned root attributes had a strong negative correlation with the leaf LDT, LBT, LCT, LUT, LCA, and LST (Fig 7B). In the case of the stem and leaf matrix, the stem attributes SCS, SCT, SPA, SCA, and SET were positively correlated with the leaf attributes LST, LTD, LBT, LUT, and LCA, but negatively correlated with LMT, LLT, LMA and LHT (Fig 7C). Among morphological traits, except few, all the traits had strong positive correlation (Fig 7D).

## Discussion

All morphological traits and yield-contributing characteristics varied significantly in *O. europaea* cultivars. In studied olive cultivars, an increase in plant height, fruit diameter, fruit weight and fruit yield was generally linked with yield capacity [33]. It has been reported that fruit size is genetic character which vary among different cultivars year wise [34]. Both fruit diameter and weight even vary within same cultivar of olive depending on genotype, cultural practices being used in the area and soil conditions like soil fertility and available moisture [27].

Morphological parameters, especially plant height and trunk circumference were strongly associated with yield-contributing traits. Moreover, fruit yield was positively correlated with leaf size, but number of leaves per branch was negatively correlated. Fruit length, stone weight, stone length, and stone diameter were not changed in all cultivars of olive. Number of leaves decreased with increasing plant height in our study, but leaf area (particularly leaf length) increased. It has been reported [35] that plant size decreases with an increase in number of leaves per branch. A decreased number of leaves in spite of an increase in leaf size was linked with reduced transpiration rate in olive cultivars, hence a major factor for controlling yield in this species [36].

Anatomical changes are more susceptible to environmental changes and exhibit a significant response to biotic and abiotic restrictions that effect overall fruit yield [37]. Sclerenchyma, collenchyma and phloem were the thickest in the roots of these woody species [38], which are directly related to increased mechanical strength, also beneficial for controlling water

**Table 3. Pearson's correlation coefficients drawn for mophological and anatomical traits of against yield attributes of *Olea europaea* L. culltivers.**

| | *FL* | *FD* | *FW* | *SW* | *SL* | *SD* | *FY* |
|---|---|---|---|---|---|---|---|
| **PH** | 0.865 | 0.708 | 0.717 | 0.817 | 0.677 | 0.730 | 0.930 |
| **TC** | 0.615 | 0.421 | 0.681 | 0.548 | 0.638 | 0.682 | 0.814 |
| **NL** | -0.690 | -0.492 | -0.587 | -0.403 | -0.111 | -0.545 | -0.724 |
| **NF** | -0.195 | -0.351 | -0.443 | -0.018 | -0.090 | 0.056 | 0.136 |
| **LL** | 0.714 | 0.610 | 0.576 | 0.547 | 0.291 | 0.475 | 0.810 |
| **LW** | 0.594 | 0.617 | 0.480 | 0.256 | -0.043 | 0.234 | 0.577 |
| **LA** | 0.652 | 0.626 | 0.517 | 0.363 | 0.052 | 0.328 | 0.676 |
| **RET** | -0.368 | -0.413 | -0.323 | 0.083 | 0.032 | -0.086 | -0.070 |
| **RCT** | -0.783 | -0.617 | -0.620 | -0.759 | -0.567 | -0.663 | -0.946 |
| **RCA** | -0.853 | -0.746 | -0.718 | -0.717 | -0.481 | -0.655 | -0.888 |
| **RST** | 0.740 | 0.540 | 0.595 | 0.497 | 0.379 | 0.606 | 0.841 |
| **ROT** | -0.909 | -0.773 | -0.827 | -0.759 | -0.592 | -0.707 | -0.959 |
| **RPT** | 0.759 | 0.589 | 0.669 | 0.795 | 0.595 | 0.719 | 0.963 |
| **RMA** | 0.685 | 0.711 | 0.631 | 0.325 | 0.141 | 0.308 | 0.646 |
| **RPA** | -0.745 | -0.524 | -0.572 | -0.610 | -0.396 | -0.686 | -0.857 |
| **RCS** | -0.779 | -0.735 | -0.628 | -0.275 | -0.245 | -0.299 | -0.546 |
| **SET** | -0.845 | -0.696 | -0.718 | -0.704 | -0.573 | -0.681 | -0.942 |
| **SCT** | -0.780 | -0.639 | -0.634 | -0.737 | -0.623 | -0.669 | -0.924 |
| **SCA** | -0.867 | -0.759 | -0.806 | -0.715 | -0.597 | -0.652 | -0.955 |
| **SST** | -0.254 | -0.195 | -0.163 | -0.151 | -0.068 | -0.137 | 0.033 |
| **SOT** | 0.846 | 0.743 | 0.766 | 0.800 | 0.542 | 0.668 | 0.947 |
| **SPT** | 0.811 | 0.676 | 0.701 | 0.792 | 0.566 | 0.736 | 0.934 |
| **SMA** | 0.661 | 0.559 | 0.546 | 0.667 | 0.454 | 0.551 | 0.879 |
| **SPA** | -0.779 | -0.599 | -0.632 | -0.723 | -0.548 | -0.686 | -0.958 |
| **SCS** | -0.565 | -0.518 | -0.346 | -0.378 | -0.487 | -0.231 | -0.676 |
| **LST** | -0.468 | -0.377 | -0.465 | -0.069 | -0.387 | -0.148 | -0.390 |
| **LPT** | 0.783 | 0.703 | 0.627 | 0.695 | 0.553 | 0.578 | 0.898 |
| **LUT** | -0.874 | -0.696 | -0.742 | -0.696 | -0.631 | -0.714 | -0.947 |
| **LDT** | -0.421 | -0.396 | -0.319 | -0.654 | -0.500 | -0.422 | -0.594 |
| **LBT** | -0.564 | -0.407 | -0.513 | -0.726 | -0.550 | -0.614 | -0.787 |
| **LMA** | 0.836 | 0.774 | 0.692 | 0.732 | 0.657 | 0.552 | 0.830 |
| **LHT** | 0.828 | 0.632 | 0.691 | 0.756 | 0.551 | 0.746 | 0.949 |
| **LCT** | -0.764 | -0.717 | -0.672 | -0.585 | -0.396 | -0.512 | -0.874 |
| **LCA** | -0.814 | -0.718 | -0.649 | -0.690 | -0.546 | -0.589 | -0.913 |
| **LMT** | 0.623 | 0.498 | 0.391 | 0.633 | 0.517 | 0.556 | 0.774 |
| **LLT** | 0.592 | 0.535 | 0.400 | 0.671 | 0.362 | 0.556 | 0.720 |

| | | |
|---|---|---|
| **Positively significant at p>0.001** | **Positively significant at p>0.01** | **Positively significant at p>0.05** |
| **Negatively significant at p>0.001** | **Negatively significant at p>0.01** | **Negatively significant at p>0.05** |

**Morphology:** PH: Plant height, TC: Trunk circumference, NL: Number of leaves per branch, NF: Number of fruits per branch (Lowest branch), LL: Leaf length, LW: Leaf width, LA: Leaf area, FL: Fruit length, FD: Fruit diameter, FW: Fruit weight, SW: Stone weight, SL: Stone length, SD: Stone diameter, FY: Fruit yield.

**Root anatomy:** RCS-Root cross sectional area, RET-Epidermal thickness, RCT-Cortical region thickness, RCA-Cortical cell area, RST-Sclerenchymatous thickness, ROT-Collenchymatous thickness, RPT-Phloem thickness, RMA-Metaxylem vessel diameter, RPA-Pith cell area.

**Stem anatomy:** SCS-Stem cross-sectional area, SET-Epidermal thickness, SCT-Cortical region thickness, SCA-Cortical cell area, SST-Sclerenchymatous thickness, SOT-Collenchymatous thickness, SPT-Phloem thickness, SMA-Metaxylem vessel diameter, SPA-Pith cell area.

**Leaf anatomy:** LMT-Midrib thickness, LLT-Lamina thickness, LST-Spongy cell thickness, LPT-Palisade cell thickness, LUT-Cuticle thickness, LDT-Adaxial epidermal thickness, LBT-Abaxial epidermal thickness, LMA-Metaxylem vessel diameter, LHT-Phloem thickness, LCT-Parenchymatous region thickness, LCA-Parenchymatous cell area.

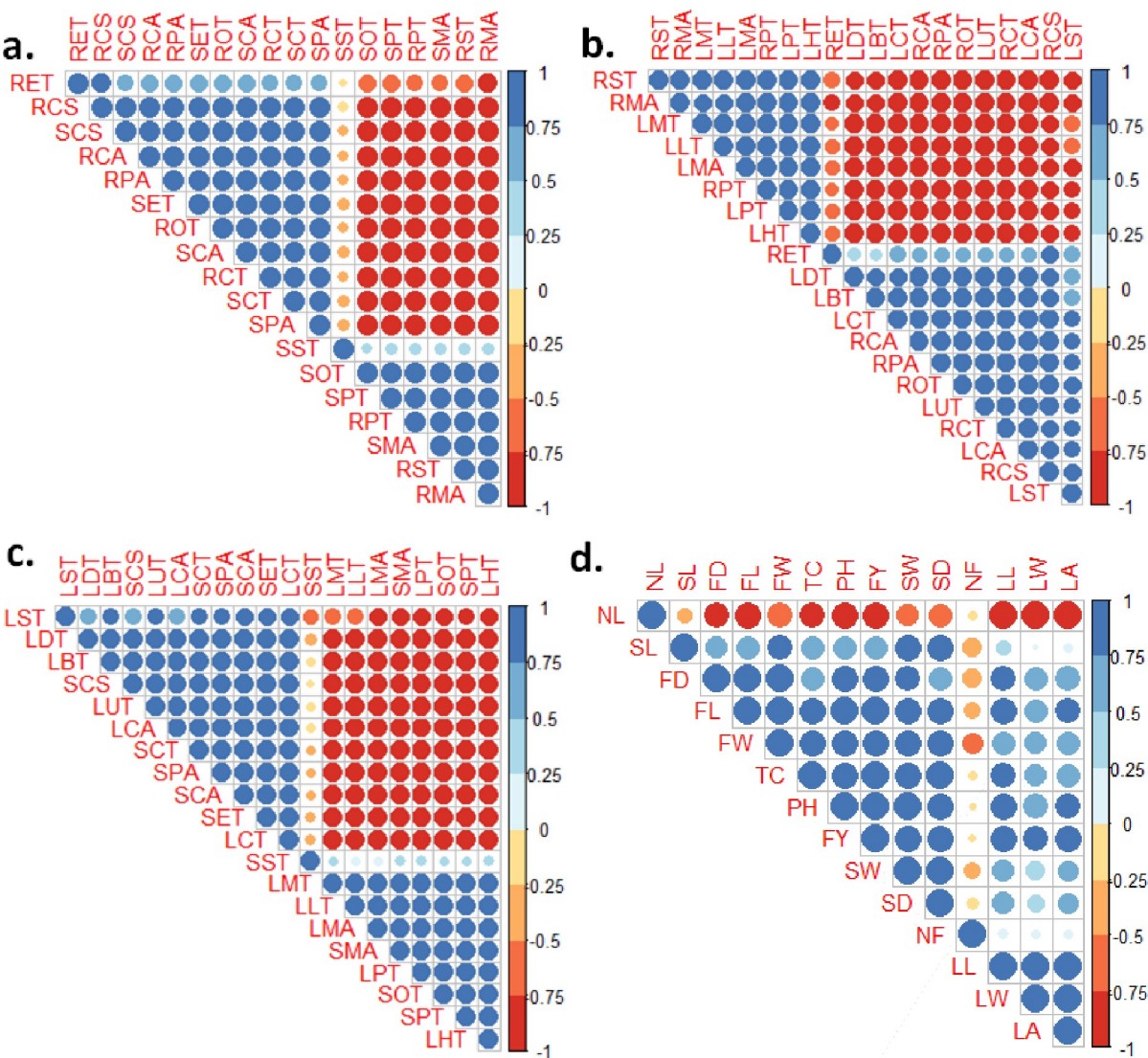

**Fig 7. Pearson's correlation coefficients between mophological and anatomical traits of *Olea europaea* L. culltivers at p>0.001, 0.01 and 0.05 levels. Legends:** a. between root and stem anatomy, b. between root and leaf anatomy, c. between stem and leaf anatomy, d. between morphological traits: RCS-Root cross sectional area, RET-Epidermal thickness, RCT-Cortical region thickness, RCA-Cortical cell area, RST-Sclerenchymatous thickness, ROT-Collenchymatous thickness, RPT-Phloem thickness, RMA-Metaxylem vessel diameter, RPA-Pith cell area. SCS-Stem cross-sectional area, SET-Epidermal thickness, SCT-Cortical region thickness, SCA-Cortical cell area, SST-Sclerenchymatous thickness, SOT-Collenchymatous thickness, SPT-Phloem thickness, SMA-Metaxylem vessel diameter, SPA-Pith cell area. LMT-Midrib thickness, LLT-Lamina thickness, LST-Spongy cell thickness, LPT-Palisade cell thickness, LUT-Cuticle thickness, LDT-Adaxial epidermal thickness, LBT-Abaxial epidermal thickness, LMA-Metaxylem vessel diameter, LHT-Phloem thickness, LCT-Parenchymatous region thickness, LCA-Parenchymatous cell area; PH: Plant height, TC: Trunk circumference, NL: Number of leaves per branch, NF: Number of fruits per branch (Lowest branch), LL: Leaf length, LW: Leaf width, LA: Leaf area, FL: Fruit length, FD: Fruit diameter, FW: Fruit weight, SW: Stone weight, SL: Stone length, SD: Stone diameter, FY: Fruit yield.

movement across the roots [38]. Cortical thickness increased the absorption capacity of roots by increasing the number of absorptive tissues [39]. In this situation, an increase in protective epidermal tissues is important for better survival under environmental adversaries [40]. Pith and root cross sectional area were not changed in the olive cultivars. Enhanced cortical region and metaxylem vessel size and pith cell area are very useful for storage as well as transport of the water that contributes to overall yield [41].

The Chakwal area is arid to semi-arid, where drought is the major factor reducing yield. Of all terrestrial plants, those from arid regions have some of the most diversified morphological and anatomical adaptations. The majority of perennial plants are xerophytes, which exhibit mechanisms that keep water from evaporating during hot conditions, offer structural support to prevent cell collapse during dry periods, or store water in tissues involved in photosynthesis [42]. By large, in studied cultivars, water stress reduces stem diameter, phloem area, and vascular bundle area while stem epidermis thickness and sclerenchyma thickness was increased. Sclerenchymatous thickness and stem cross-sectional area did not changed in different olive cultivars. Sclerification in cortical region is linked with greater root strength and root tip bending capacity [43]. It also improves root depth, plant growth and increase root penetration ability into compact soil [44]. Extensive sclerification in epidermal cells may have a substantial role in reducing water loss in water deficit conditions [37]. Increased phloem thickness was a critical adaptation that aids in sap transport [45]. Collenchymatous thickness was important in providing mechanical support [30]. Metaxylem vessel diameter is very helpful for survival of plants for a long period [15]. Increasing meta-xylem area is crucial for the growth of cortical parenchyma and can be directly related to the conduction of water and minerals [46]. Water conductivity in the xylem is influenced by xylem diameter. It has been proposed that an enlarged xylem channel size is a beneficial characteristic for enhancing water absorption from deeper soil layers [47]. Increased proportion of parenchymatous cells in pith and cortical region in olive cultivars is associated with stress tolerance as it maintains solute conduction as well as storage, which directly contributes towards yield stability [48]. It has been demonstrated that cortical cell size and composition are crucial in trees for soil penetration [49]. Mechanical resistance is influenced by a number of physical soil factors, including water content, texture, and bulk density. These characteristics can affect root elongation, crop growth, and ultimately plant yield [50].

The leaf is the most adaptable organ in its response to environmental conditions [18]. Leaf structures reflect the effects of water stress more clearly than those of stems or roots [51]. Leaf thickness is a quantitative feature linked to the plant's ability to thrive in dry, high irradiance [52]. Plants are benefitted from leaf modifications such as increase in the epidermis thickness (adaxial and abaxial), which works as a first line of defense against environmental stress and as a means of self-preservation in plants [39]. Leaf thickness, as well as the distinction of palisade and spongy mesophyll may have a direct impact on light uptake [34]. Cuticle, spongy mesophyll and palisade mesophyll varied significantly in the olive cultivars, while epidermal thickness remained unchanged on both leaf surfaces. The plant cuticle is the last barrier for water to pass through [40,53]. Increased leaf thickness among cultivars may be linked to the maintenance of higher leaf water contents, facilitate leaf hydraulic conductivity, and water storage capacity under water limitations [54,55].

Leaf midrib thickness in olive cultivars seems depended upon proportion of vascular as well as parenchymatous regions, therefore thicker midrib is the indication of better conduction of photoassimilates and storage [56]. The sap flow velocity is controlled by phloem and related with xylem for exchange of water and carbohydrates [57]. Proportion of parenchymatous region in leaves provides large space for absorption of water and main storage compartment for sugars and other solutes in mature stem and leaves [41]. Reduced size of metaxylem vessels is a major anatomical adaptation under stressful conditions like drought, as it protects cavitation in the vessels and prevents vessels to collapse and therefore directly involve in yield stability [58].

Improving yield capacity by increasing plant height, number of leaves per branch, number of fruits per branch, fruit diameter, fruit weight, and fruit yield was deemed one of the most important strategies of olive cultivars. Anatomical changes at the root, stem, and leaf levels

were important because these were influenced by environmental adversaries. The increase in protective epidermal tissues was critical for better survival under extreme condition, while cortical thickness enhanced the absorption capacity of roots under drought. In stem adaptations, collenchymatous thickness, phloem thickness and metaxylem area was strongly related to yield contributing traits. The palisade and spongy mesophylls and leaf thickness had a direct impact on light intake. Plant cuticle (and epidermis) acted as a barrier for the movement of water.

## Conclusion

It is concluded that all studied morphological characters, yield and yield parameters, and root, stem and leaf anatomical characteristics varied highly significantly in olive cultivars grown at Barani Agricultural Research Institute, Chakwal. The most promising cultivar regarding yield was Erlik, in which morphological features like plant height, stem circumference and yield contributor seed weight were the maximum. Root anatomical characteristics like epidermal thickness and phloem thickness, stem features like collenchymatous thickness, phloem thickness and metaxylem vessel diameter, and leaf traits like midrib thickness, palisade cell thickness a phloem thickness were the highest. The second best Hamdi showed the maximum value for plant height and yield contributing traits like fruit length, weight and diameter and seed length and weight. Stem phloem thickness, and leaf anatomical characteristics like midrib and lamina thicknesses, palisade cell thickness, metaxylem vessel diameter and phloem thickness were the maximum. Fruit yield in the studied olive cultivars can be more closely linked to high proportion of storage parenchyma, broader xylem vessels and phloem proportion, dermal tissue and high proportion of collenchyma. Anatomically compact parenchyma tissue and more number of collenchyma layers offer drought resistance.

## Acknowledgments

This manuscript has been derived from MPhil Thesis of the second author submitted to University of Sargodha, Sargodha

## Availability of data and material

The voucher specimens used for plant identification are deposited to the herbarium facility of the Department of Botany, University of Agriculture, Faisalabad, and are available for verification on request. The minimal data set underlying the results described in manuscript, anatomical slides, photographs and raw data calculated from these photographs are available with primary author and can be requested if needed.

## Code availability

R codes and modeling details are available with author(s) listed as bio-statisticians under author's contribution section of declarations and can be requested if needed to reproduce the data visualization or other results.

## Publication ethics statements

It is certified that the manuscript is the product of an original study and is submitted solely to this Journal for consideration. It is not submitted to any other Journal, in part or full, for simultaneous consideration nor has been previously published in any form or language (other than as a thesis of the first author, which is properly acknowledged). There is no plagiarism/ self-plagiarism, salami-slicing/publishing, secondary publication nor near verbatim. All data presented in this manuscript is product of our own study and the manuscript does not contain

any copyrighted material (data tables or figures). All results and data are presented clearly, honestly, and without fabrication, falsification or inappropriate data manipulation (including image-based manipulation).

## Author Contributions

**Conceptualization:** Iftikhar Ahmad, Mansoor Hameed.

**Data curation:** Sana Fatima, Sana Basharat, Ansa Asghar, Syed Mohsan Raza Shah.

**Formal analysis:** Sana Fatima, Muhammad Sajid Aqeel Ahmad, Farooq Ahmad, Ansar Mehmood.

**Methodology:** Muhammad Sajid Aqeel Ahmad, Farooq Ahmad, Syed Mohsan Raza Shah.

**Resources:** Farooq Ahmad.

**Software:** Muhammad Sajid Aqeel Ahmad, Ansar Mehmood.

**Supervision:** Mohammad Sohail, Mansoor Hameed.

**Validation:** Ansar Mehmood.

**Visualization:** Syed Mohsan Raza Shah.

**Writing – review & editing:** Khawaja Shafique Ahmad.

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
