## [Decision Letter · Decision Letter 0]

23 Feb 2023

PONE-D-22-35252

Morpho-anatomical determinants of yield potential in Olea europaea L. cultivars belonging to diversified origin grown in semi-arid environments

PLOS ONE

Dear Dr. Ahmad,

Thank you for submitting your manuscript to PLOS ONE. After careful consideration, we feel that it has merit but does not fully meet PLOS ONE’s publication criteria as it currently stands. Therefore, we invite you to submit a revised version of the manuscript that addresses the points raised during the review process.

We look forward to receiving your revised manuscript.

Kind regards,

Rupesh Kailasrao Deshmukh, Ph.D.

Academic Editor

PLOS ONE

Journal Requirements:

4. Please ensure that you refer to Figures 1-4 in your text as, if accepted, production will need this reference to link the reader to the figure.

Reviewers' comments:

Reviewer's Responses to Questions

**Comments to the Author**

1. Is the manuscript technically sound, and do the data support the conclusions?

Reviewer #1: Yes

Reviewer #2: Yes

2. Has the statistical analysis been performed appropriately and rigorously? 

Reviewer #1: Yes

Reviewer #2: Yes

3. Have the authors made all data underlying the findings in their manuscript fully available?

Reviewer #1: Yes

Reviewer #2: Yes

4. Is the manuscript presented in an intelligible fashion and written in standard English?

Reviewer #1: No

Reviewer #2: Yes

5. Review Comments to the Author

Reviewer #1: The discussion should be enriched and the main text should be edited by a native speaker or professional editing service. The data of this study would be valuable to the related research field and agricultural practice. The authors should review and discuss more related references for this manuscript, maybe 10-20 up-to-date publications can be added to the section of references.

Reviewer #2: This manuscript attempts to characterize some traits of olive cultivars in Pakistan. The works seems to be original, but some improvements need to be done before is suitable for publication.

In the future, please, include numbers in every line of the manuscript for easier review.

Introduction

Paragraph starting by “Olive (Olea europaea L.) is an important oil yielding, medium sized woody and evergreen tree species growing in the Mediterranean region. It has approximately 25 genera and 600 species…” should be placed before mentioning any other work on the olive species.

The sentence “It was hypothesized that internal structure being the main contributor of overall growth should also be contributory elements to yield of studied cultivars” is not fully clear to me. Please, rephrase.

Materials and methods

Manzanilla is a Spanish cultivar and Fs-17 comes from an Italian breeding program.

Subheadings “Experimental conditions and layout of the experiment” and “Growth and yield measurements” should be merged and clarify. First, you have to present the orchard sampled, including the date of plating (2010 I think). And then, describing the sampling strategy. Particularly, it is not clear the way that the yield has been measured. It is compulsory to have a measure of the yield of the whole tree in order to be able to associate with morphological and anatomical traits of the plant.

Considering the sentence “To investigate the anatomical features, olive plants’ root, stem and leaf were separated”, it seems that the full plant was destroyed to gather the root and shoots data, but this is not fully clear. It should also be stated how many branches per tree were analyzed. And if the age of the (one year old, probably if they have fruits). Besides it is not clear how yield was measured. In fact, the sentence “Young stems were collected near the branch apex” it is not clear, and a more extended explanation is needed.

Results

You have to give some data on the environmental conditions of the study site (maximum, minimum and average daily temperature per month and rainfall) and if there is irrigation, indicate the yearly amount of water applied.

At the beginning of Results section, it is indicated that “Plant height did not show any significant difference, which was the maximum in three cultivars Erlik, Hamdi and BARI-2, while the minimum (4.4 m) in the HP Olive and QR Olive cultivars. Trunk circumference showed no significant difference in all cultivars.” However, in Table 1, significant differences for both plant height and trunk circumference are indicated. Please, clarify this and review the rest of the comments of the results section for similar contradictions.

Figures 1 to 4 are not mentioned in text. You should either include in the text or eliminate them.

It would be interesting to know the correlations among morphological and anatomical traits themselves.

6. PLOS authors have the option to publish the peer review history of their article (what does this mean?). If published, this will include your full peer review and any attached files.

Reviewer #1: No

Reviewer #2: No

---

## [Author Response · Author response to Decision Letter 0]

12 Apr 2023

PONE-D-22-35252

Morpho-anatomical determinants of yield potential in Olea europaea L. cultivars belonging to diversified origin grown in semi-arid environments.

We sincerely thank the editor and anonymous reviewers for taking the time to review our manuscript and providing constructive feedback to improve our manuscript. We have revised the manuscript accordingly by following the reviewers' suggestion. A detailed response of each comment is apprehended below and hope that the correction will meet with approval. 

Editorial comments

Response: We have ensured that your manuscript meets PLOS ONE's style requirements, including those for file naming.

We suggest you thoroughly copyedit your manuscript for language usage, spelling, and grammar. 

Response: During review we have thoroughly copyedited the manuscript for language usage, spelling and grammar mistakes. 

In your Data Availability statement, you have not specified where the minimal data set underlying the results described in your manuscript can be found.

Response: Statement has been revised and provided as “the minimal data set underlying the results described in manuscript, anatomical slides, photographs and raw data calculated from these photographs are available with primary author and can be requested if needed’’.

Please ensure that you refer to Figures 1-4 in your text as, if accepted, production will need this reference to link the reader to the figure.

Response: We have ensured that all the tables and figures are cited in the main text of the manuscript. 

We note that Figure 1 in your submission contain [map/satellite] images which may be copyrighted. 

Response: We found figure 1 unnecessary, so it has been deleted. 

Reviewer #1 

The discussion should be enriched and the main text should be edited by a native speaker or professional editing service. The data of this study would be valuable to the related research field and agricultural practice. The authors should review and discuss more related references for this manuscript, maybe 10-20 up-to-date publications can be added to the section of references.

Response: We sincerely appreciate all the valuable comments and suggestions, which really helped us to improve the quality of the manuscript. As per your suggestions, we have revised the discussion part and the main text of the manuscript has been edited. Additionally, we have updated the reference list. 

Reviewer #2

This manuscript attempts to characterize some traits of olive cultivars in Pakistan. The works seems to be original, but some improvements need to be done before is suitable for publication.

Response: We can’t thank you enough for taking the time to review our manuscript and providing constructive comments on our work which really helped us in revision process. We have revised the manuscript in the light of your comments. 

In the future, please, include numbers in every line of the manuscript for easier review.

Response: Line number of the manuscript have been added. 

Introduction

Paragraph starting by “Olive (Olea europaea L.) is an important oil yielding, medium sized woody and evergreen tree species growing in the Mediterranean region. It has approximately 25 genera and 600 species…” should be placed before mentioning any other work on the olive species.

Response: Line 73-80. We have placed this paragraph before the other work reported on Olea europaea. 

The sentence “It was hypothesized that internal structure being the main contributor of overall growth should also be contributory elements to yield of studied cultivars” is not fully clear to me. Please, rephrase.

Response: The sentence has been rephrased. 

Materials and methods

Manzanilla is a Spanish cultivar and Fs-17 comes from an Italian breeding program.

Response: Sorry for mistakes. Corrections have been made. 

Subheadings “Experimental conditions and layout of the experiment” and “Growth and yield measurements” should be merged and clarify. First, you have to present the orchard sampled, including the date of plating (2010 I think). And then, describing the sampling strategy. Particularly, it is not clear the way that the yield has been measured. It is compulsory to have a measure of the yield of the whole tree in order to be able to associate with morphological and anatomical traits of the plant.

Response: Subheadings “Experimental conditions and layout of the experiment” and “Growth and yield measurements” have been merged and clarified. Information have been provided and it is now clear how yield and associated morphological and anatomical traits were measured. 

Considering the sentence “To investigate the anatomical features, olive plants’ root, stem and leaf were separated”, it seems that the full plant was destroyed to gather the root and shoots data, but this is not fully clear. It should also be stated how many branches per tree were analyzed. And if the age of the (one year old, probably if they have fruits). Besides it is not clear how yield was measured. In fact, the sentence “Young stems were collected near the branch apex” it is not clear, and a more extended explanation is needed.

Response: Sentence has been modified for clarity and more details are given in M&M section. Root and shoot data collected from one year old fruited cultivars without damaging full tree. For average yield per plant, fruit was harvested manually and total fruit was weighed for each year was calculated.

Results

You have to give some data on the environmental conditions of the study site (maximum, minimum and average daily temperature per month and rainfall) and if there is irrigation, indicate the yearly amount of water applied.

Response: Data on the environmental conditions of the study site is provided in M&M section. 

At the beginning of Results section, it is indicated that “Plant height did not show any significant difference, which was the maximum in three cultivars Erlik, Hamdi and BARI-2, while the minimum (4.4 m) in the HP Olive and QR Olive cultivars. Trunk circumference showed no significant difference in all cultivars.” However, in Table 1, significant differences for both plant height and trunk circumference are indicated. Please, clarify this and review the rest of the comments of the results section for similar contradictions.

Response: Results have been checked and revised for similar contradictions. 

Figures 1 to 4 are not mentioned in text. You should either include in the text or eliminate them. It would be interesting to know the correlations among morphological and anatomical traits themselves.

Response: Figures 1 to 4 are now cited in the main body text. Correlations among morphological and anatomical traits is provided in figure 7.

---

## [Decision Letter · Decision Letter 1]

22 May 2023

Morpho-anatomical determinants of yield potential in Olea europaea L. cultivars belonging to diversified origin grown in semi-arid environments

PONE-D-22-35252R1

Dear Dr. Ahmad,

We’re pleased to inform you that your manuscript has been judged scientifically suitable for publication and will be formally accepted for publication once it meets all outstanding technical requirements.

Kind regards,

Rupesh Kailasrao Deshmukh, Ph.D.

Academic Editor

PLOS ONE

Additional Editor Comments (optional):

Reviewers' comments:

Reviewer's Responses to Questions

**Comments to the Author**

1. If the authors have adequately addressed your comments raised in a previous round of review and you feel that this manuscript is now acceptable for publication, you may indicate that here to bypass the “Comments to the Author” section, enter your conflict of interest statement in the “Confidential to Editor” section, and submit your "Accept" recommendation.

Reviewer #2: All comments have been addressed

Reviewer #3: All comments have been addressed

2. Is the manuscript technically sound, and do the data support the conclusions?

Reviewer #2: Yes

Reviewer #3: Yes

3. Has the statistical analysis been performed appropriately and rigorously? 

Reviewer #2: Yes

Reviewer #3: Yes

4. Have the authors made all data underlying the findings in their manuscript fully available?

Reviewer #2: Yes

Reviewer #3: Yes

5. Is the manuscript presented in an intelligible fashion and written in standard English?

Reviewer #2: Yes

Reviewer #3: Yes

6. Review Comments to the Author

Reviewer #2: In my opinion, the manuscript is suitable for publication in its present form. The manuscript is original and all the suggested changes has been incorporated.

Reviewer #3: The authors have addressed all the questions of the reviewers and the manuscript has been improved from the previous version. The present version looks appropriate to be considered for publication.

7. PLOS authors have the option to publish the peer review history of their article (what does this mean?). If published, this will include your full peer review and any attached files.

Reviewer #2: **Yes: **Raul de la Rosa

Reviewer #3: No

---

## [Editor Report · Acceptance letter]

26 May 2023

PONE-D-22-35252R1 

Morpho-anatomical determinants of yield potential in *Olea europaea* L. cultivars belonging to diversified origin grown in semi-arid environments 

Dear Dr. Ahmad:

I'm pleased to inform you that your manuscript has been deemed suitable for publication in PLOS ONE. Congratulations! Your manuscript is now with our production department. 

Kind regards, 

on behalf of

Dr. Rupesh Kailasrao Deshmukh 

Academic Editor

PLOS ONE